# Recent Progress of Antisense Oligonucleotide Therapy for *Superoxide-Dismutase-1*-Mutated Amyotrophic Lateral Sclerosis: Focus on Tofersen

**DOI:** 10.3390/genes15101342

**Published:** 2024-10-20

**Authors:** Hidenori Moriyama, Toshifumi Yokota

**Affiliations:** 1Department of Medical Genetics, Faculty of Medicine and Dentistry, University of Alberta, Edmonton, AB T6G 2H7, Canada; 2The Friends of Garrett Cumming Research & Muscular Dystrophy Canada HM Toupin Neurological Sciences Research, Edmonton, AB T6G 2H7, Canada

**Keywords:** amyotrophic lateral sclerosis (ALS), *superoxide dismutase 1* (*SOD1*), antisense oligonucleotide (ASO), tofersen, central nervous system (CNS)

## Abstract

Amyotrophic lateral sclerosis (ALS) is a refractory neurodegenerative disease characterized by the degeneration and loss of motor neurons, typically resulting in death within five years of onset. There have been few effective treatments, making the development of robust therapies an urgent challenge. Genetic mutations have been identified as contributors to ALS, with mutations in *superoxide dismutase 1* (*SOD1*), which neutralizes the harmful reactive oxygen species superoxide, accounting for approximately 2% of all ALS cases. To counteract the toxic gain of function caused by *SOD1* mutations, therapeutic strategies aimed at suppressing *SOD1* gene expression have shown promise. Antisense oligonucleotide (ASO) is an artificially synthesized, short, single-stranded DNA/RNA molecule that binds to target RNA to alter gene expression, representing a next-generation therapeutic approach. In 2023, tofersen became the first ASO drug approved by the FDA for ALS. Administered intrathecally, tofersen specifically binds to *SOD1* mRNA, inhibiting the production of toxic SOD1 protein, thereby improving biomarkers of ALS. The long-term efficacy and safety of tofersen require further validation, and the development of more optimized treatment protocols is essential. A series of studies and therapeutic developments related to *SOD1* mutations have advanced the understanding of ALS pathophysiology and significantly contributed to treatment strategies for central nervous system disorders. This review focuses on an overview of *SOD1* mutations and the development process of tofersen, aiming to deepen the understanding of advancements in ALS research and discuss future challenges and directions for ASO therapy.

## 1. Introduction

Amyotrophic lateral sclerosis (ALS) is one of the neurodegenerative diseases first described by Jean-Martin Charcot in 1869 [1]. ALS is a relatively rare disease with an incidence rate of about 2 per 100,000 person-years worldwide [2,3]. It is characterized by the gradual degeneration and loss of both upper and lower motor neurons, leading to voluntary muscle atrophy, paralysis, and ultimately death within two to five years due to swallowing and respiratory muscle paralysis [4,5]. In addition to motor neuron damage, ALS is frequently associated with dementia, and there is accumulating evidence that the disease spectrum is identical to that of frontotemporal dementia (FTD) [6]. Advances in the development and improvement of respiratory assist devices, as well as the emergence of enteral nutrition and gastrostomy, have enabled longer-term care than in the past. However, the underlying cause of motor neuron degeneration remains completely unknown even 150 years after the initial description of the disease, and no fundamental treatment based on its etiology has been established.

Since the identification of *superoxide dismutase 1* (*SOD1*) as a cause of familial ALS in 1993 [7], the significance of genetic mutations in ALS has become increasingly clear. Consequently, there has been a surge in the exploration of therapeutic strategies that target specific DNA and RNA as novel treatment approaches for ALS. The development of antisense oligonucleotide (ASO) therapies for *SOD1*-mutation-related ALS has progressed through various preclinical and clinical trials, culminating in the Food and Drug Administration (FDA)’s approval of tofersen in April 2023 [8].

This review aims to deepen the understanding of the role of genetic mutations in ALS and the development of next-generation therapies, focusing specifically on *SOD1* mutations and the journey of tofersen. We also discuss the future directions of ASO in ALS.

## 2. Genetic Background of ALS

### 2.1. Overview

About 10% of ALS patients have a family history of the disease, and to date, more than 30 genes associated with familial cases of ALS have been identified [2,3,9]. Notably, many of the mutated gene products are involved in RNA metabolism, intracellular transport, or protein degradation mechanisms [10]. The remaining 90% of cases are sporadic; however, the pathogenic mutations that have been identified in familial ALS have also been observed in many sporadic cases [11]. Genetic research, including family aggregation studies, twin studies, and genome-wide association studies (GWAS), has shown that ALS has a significant genetic component, with estimates of 61% from twin studies and 21% from GWAS [12,13,14]. Four representative ALS-related genes are *chromosome 9 open reading frame 72* (*C9orf72*), *SOD1*, *fused in sarcoma* (*FUS*), and *TAR DNA-binding protein* (*TARDBP*). These pathogenic mutations account for about 60% of familial cases and about 10% of sporadic cases of ALS [2]. Additionally, methods such as whole exome sequencing have led to the discovery of less frequent genetic mutations, identifying genetic mutations in 76% of familial cases and 25% of sporadic cases [2,10,15]. Several gene mutations, including *C9orf72* and *TARDBP* mutations, are also known to cause FTD [16]. Moreover, environmental and lifestyle factors also contribute to ALS, in addition to genetic mutations [17]. For instance, alcohol consumption, educational level, physical activity, dyslipidemia, and smoking have been identified as risk factors for ALS and are thought to interact with the genetic background to influence the onset of the disease [17,18,19,20].

### 2.2. ALS with SOD1 Mutations

The *SOD1* gene was first identified as a causative gene for ALS in 1993 [7]. This finding was significant, as it was the first to show that genetic abnormalities play a role in the development of ALS. *SOD1* encodes a protein of 154 amino acids that acts as an enzyme, catalyzing the breakdown of superoxide radicals into oxygen and hydrogen peroxide [21]. To date, over 200 mutations in the *SOD1* gene have been reported (Figure 1). In Europe, *SOD1* mutations are the second most prevalent cause of ALS after *C9orf72* mutations, accounting for 12–15% of familial ALS and 1–2% of sporadic ALS cases [16,22,23]. In Japan, *SOD1* mutations are the most prevalent, representing about 30% of familial ALS and 1–2% of sporadic ALS cases [16,24]. Most pathogenic *SOD1* mutations are missense mutations [22]. These mutations typically follow an autosomal dominant pattern with high penetrance, though some mutations can have asymptomatic carriers [24]. The most common clinical presentation in patients with *SOD1* mutations is leg weakness with prominent lower motor neuron signs [25]. Certain *SOD1* mutations are associated with distinct clinical phenotypes of ALS, which is important for developing personalized treatment strategies [26,27,28]. For instance, the p.A5V mutation prevalent in ALS patients in North America results in a rapidly progressive, dominant form of the disease [29,30]. Conversely, the p.H47R mutation, more common in Japan, results in a slowly progressive form [31].

The precise mechanism by which *SOD1* mutations cause toxicity is not yet fully understood, but it is generally accepted that these mutations involve a toxic gain of function [21,32,33]. Just one year after the identification of *SOD1* as an ALS causative gene, transgenic mice carrying the mutated *SOD1* gene were developed [34]. The most widely used transgenic mouse to date is the *SOD1*^G97A^ transgenic mouse, but others are known, including *SOD1*^G85R^, *SOD1*^G37R^, and *SOD1*^H46R^ transgenic mice [35]. These transgenic mice have significantly advanced our understanding of the mechanisms underlying ALS pathogenesis due to *SOD1* mutations. They exhibit disease features similar to those of ALS patients, such as muscle atrophy, glial cell activation, and motor neuron loss in the spinal cord, which has made them valuable model organisms for ALS research [34,36,37]. *SOD1* mutations are thought to cause neuronal cell death through multiple mechanisms, including oxidative stress, glutamate excitotoxicity, non-cell-autonomous glial cell toxicity, endoplasmic reticulum (ER) stress, mitochondrial dysfunction, axonal transport defects, and extracellular toxicity of mutant SOD1 proteins [38,39,40,41]. Interestingly, while the loss of *SOD1* causes motor neuron dysfunction, it does not cause motor neuron death [42,43].

## 3. Treatment of ALS

There is currently no cure for ALS. Prior to the approval of tofersen, the available treatments were limited to riluzole, a glutamate antagonist, and edaravone, a free radical scavenger (Table 1). Riluzole inhibits the release of glutamate from neurons, partly by inactivating voltage-dependent sodium channels on glutamatergic nerve terminals, and also blocks some of the postsynaptic effects of glutamate through noncompetitive blockade of N-methyl-D-aspartate (NMDA) receptors [44]. While riluzole has been shown to modestly extend survival in Phase 3 trials and subsequent population studies, its effect remains limited [45,46,47]. The most common adverse effects of riluzole include diarrhea, nausea, dizziness, fatigue, and hepatic dysfunction [41]. Edaravone, administered intravenously or orally, functions as an antioxidant. The initial randomized, double-blind, placebo-controlled Phase 3 study did not show significant efficacy for edaravone [48]. However, post hoc analyses suggested a potential benefit in a specific ALS subgroup with a shorter disease duration [49]. Consequently, additional Phase 3 trials focused on a more specific subgroup of patients with a shorter disease duration. These trials demonstrated a modest but statistically significant positive effect of edaravone on the rate of functional decline over six months, although there was no significant impact on respiratory function [50]. As a result, edaravone received FDA approval for intravenous treatment in 2017 and for oral administration in 2022. The most common adverse effects of edaravone include bruising and gait disturbances, although it is generally considered safe for use [51].

In 2022, the FDA approved a combination of sodium phenylbutyrate and taurursodiol (PB/TURSO) [52]. PB/TURSO is an oral drug with potential neuroprotective effects, believed to inhibit motor neuron apoptosis by reducing oxygen free radical production, decreasing ER stress, and inhibiting caspase activation [53]. However, this medication was voluntarily removed from the market based on topline results from the Phase 3 PHOENIX trial (NCT05021536). Research into more effective ALS drugs is ongoing, with current focuses on a variety of approaches including antioxidants, cell therapy, mitochondrial dysfunction, neuroinflammation, and proteostasis [54].

In recent years, the development of molecular therapies targeting specific genes has advanced. It is known that most monogenic causes of ALS operate through a toxic gain of function associated with the mutated gene. In this regard, ASO therapies, which aim to directly target and modify the genes responsible for the disease while neutralizing their toxic products, show considerable promise [55]. Currently, antisense therapies targeting the transcripts of *SOD1*, *FUS*, and *C9orf72* are in various stages of development [54].

## 4. ASO in CNS Disorders

### 4.1. ASO Overview

As previously mentioned, considering that the pathogenesis of *SOD1* ALS involves a dominant gain of function, therapies aimed at reducing the expression of the *SOD1* gene are promising options. Among the various methods for gene silencing, ASO therapy is currently the most promising approach (Figure 2). ASOs are single-stranded oligonucleotides, typically composed of 14 to 25 synthetic nucleotides. They bind specifically to target mRNA by Watson–Crick hybridization, with a minimal amount of off-target pairing [56,57]. Once ASOs bind to the target RNA, they modify gene expression primarily through two mechanisms: RNase H-mediated degradation of the target RNA and steric blocking of specific sites on the mRNA or pre-mRNA, which can prevent translation or modulate splicing [58]. RNase H is an enzyme present in all eukaryotic cells. It hydrolyzes RNA bound to DNA during the process of DNA replication [59]. By having a complementary sequence to the target RNA, ASOs utilize the RNA degradation capability of RNase H. When an RNA-ASO heteroduplex forms, RNase H can attach to the RNA and degrade it, thereby preventing the production of the disease-associated protein [60]. However, if all of the DNA bases within the ASO are chemically altered, RNase H cannot recognize the DNA-RNA heteroduplex, which inhibits its enzymatic function [61,62]. In the absence of RNase H activity, ASOs that bind to the target RNA can obstruct processes involved in splicing or translation (steric block). The most common application of steric block ASOs is the modulation of selective splicing, allowing for the selective exclusion or inclusion of specific exons [58]. The first ASO drug approved by the FDA for motor neuron disease is nusinersen, used for spinal muscular atrophy (SMA). Nusinersen binds to intron 7 of the *survival motor neuron 2* (*SMN2*) pre-mRNA, suppressing the skipping of exon 7, thereby producing *SMN2* mRNA that includes exon 7 and facilitating the expression of full-length SMN protein [63].

### 4.2. Challenges of ASO Therapy for CNS Disorders

A significant issue, not limited to ASOs, when delivering drugs to the central nervous system (CNS) is the presence of the blood–brain barrier (BBB). The BBB represents a significant delivery challenge to the CNS due to continuous basement membranes, dense junctional proteins, and nonfenestrated capillaries, making the extravasation of large molecules into the brain highly difficult [64]. Molecules that can pass through the BBB need to be hydrophobic, low-molecular-weight (below 500 Da) compounds [65]. Thus, how to deliver hydrophilic, high-molecular-weight ASOs to the CNS is an area of ongoing research. Intrathecal administration is a method that does not require consideration of the BBB, allowing for high efficacy in delivering ASOs to the CNS. Studies in rodents and primates have shown that intrathecal delivery of ASOs results in wide distribution and tissue penetration throughout the CNS [66,67,68]. It is common to administer ASOs alone without the aid of delivery molecules. However, a major drawback of this method is its invasiveness. In clinical trials, side effects such as headaches and back pain from lumbar punctures have been frequently reported, which may hinder treatment continuation or necessitate schedule changes [69,70].

Unfortunately, there is currently no established method for delivering ASOs to the CNS via alternative routes that can cross the BBB, but advancements in drug delivery technologies are underway [71]. For example, cell-penetrating peptides that can bind to ASOs may provide a promising strategy by facilitating the delivery of ASOs across the BBB, thereby bypassing the need for invasive intrathecal injections and the associated side effects, which could potentially improve treatment efficacy for CNS disorders [57].

## 5. ASO Therapy for *SOD1* ALS

### 5.1. Background Leading to the Development of Tofersen

ASO therapy targeting *SOD1* has been explored for nearly 20 years. ASO333611 is a 2′-O-Methoxyethyl (2′-MOE) gapmer against *SOD1* and was developed as the first ASO therapy for the CNS [66]. Initial studies demonstrated that when ASOs are administered intrathecally, they distribute widely throughout the CNS and penetrate tissues effectively. Using transgenic rats expressing human mutant *SOD1*^G93A^ gene, ASO333611 was continuously infused at a dose of 100 µg/day into the lateral ventricle via an osmotic pump starting at postnatal day 65, approximately 30 days before the expected onset of disease [72]. This resulted in a significant reduction in *SOD1*^G93A^ mRNA levels in the brainstem and across all levels of the spinal cord, as well as a significant decrease in SOD1^G93A^ protein levels in the cervical cord. Although there was no observed slowing of early disease progression, a 37% increase in survival was noted.

These encouraging preclinical findings led to a Phase 1, double-blind, placebo-controlled clinical trial (NCT01041222) aimed at assessing the safety and tolerability of intrathecally administered ASO333611 in patients with *SOD1* ALS [73]. Thirty-two ALS patients with *SOD1* mutations were enrolled, including eight in the placebo group. Low doses of ASO333611 (0.15–3 mg) were administered via intrathecal infusion over 11.5 h. Adverse events were reported in 88% patients in the placebo group and 83% in the ASO333611 group. The most common events included postlumbar puncture syndrome (38% vs. 33%), back pain (50% vs. 17%), and nausea (0% vs. 13%). No dose-limiting toxic effects or any safety or tolerability concerns related to ASO333611 were observed. Because of the low dose administered, a reduction in SOD1 protein levels was not demonstrated. This trial was the first-in-man study of intrathecal ASO administration, confirming the safety and tolerability of ASO333611.

The series of preclinical data and Phase 1 trials for *SOD1* ASO have collectively demonstrated the safety of intrathecal administration and its efficacy in rodents, established protocols and evaluation methods for intrathecal administration studies, and indicated the feasibility of ASO therapy for CNS disorders [74].

### 5.2. Preclinical Study of Tofersen

Tofersen (BIIB067) is an ASO that is developed by Ionis Pharmaceuticals and is licensed to Biogen. As mentioned, the feasibility of *SOD1* ASO therapy has been suggested; however, the effects of ASO333611 were modest. With advancements in ASO development technologies over time, there was a demand for more effective *SOD1* ASOs, leading to the progression of next-generation therapeutics.

In the research, more than 2000 ASOs targeting the human *SOD1* gene were screened in vitro, followed by optimization, resulting in the selection of a potent 2′-MOE mixed-backbone ASO targeting the 3’ UTR [75]. In a human neuroblastoma cell line (SH-SY5Y), BIIB067 demonstrated a dose-dependent reduction in *SOD1* mRNA levels, showing significant effects compared with ASO333611. In vivo experiments involved bolus injections of BIIB067 into the ventricles of *SOD1*^G93A^ transgenic mice to deliver it into the cerebrospinal fluid (CSF). BIIB067 reduced human *SOD1* mRNA levels in a dose-dependent manner, achieving reduction of up to approximately 75% after a single injection. BIIB067 was significantly more active and effective than ASO333611. Similarly, in experiments where BIIB067 was bolus injected into the intrathecal space of *SOD1*^G93A^ rats, a dose-dependent decrease in *SOD1* mRNA was observed, demonstrating higher activity than ASO333611. Sustained reductions in *SOD1* mRNA were observed for about eight weeks after a single bolus administration. Furthermore, administration of BIIB067 before disease onset in *SOD1*^G93A^ mice and rats maintained motor function and significantly prolonged the time until weight loss occurred. The *SOD1* ASO extended the lifespan of *SOD1*^G93A^ rats by more than 50 days (32%) and that of *SOD1*^G93A^ mice by approximately 40 days (22%). The initial loss of compound muscle action potential in *SOD1*^G93A^ mice was reversed following a single dose of tofersen. In addition, increases in serum phosphorylated neurofilament heavy chain (pNFH), a cytoskeletal protein known to be released into the CSF and serum during axonal injury and correlated with disease severity, were ameliorated by the ASO treatment [76,77]. Intrathecal injection of BIIB067 in nonhuman primates (cynomolgus monkeys) resulted in a dose-dependent decrease in *SOD1* mRNA and protein levels in CSF and CNS tissues.

### 5.3. Clinical Study of Tofersen

#### 5.3.1. Phase 1/2 VALOR

Based on the groundbreaking effects observed in rodents, a randomized, double-blind, placebo-controlled Phase 1/2 study (VALOR; NCT02623699) was launched to assess the pharmacokinetics and tolerability of intrathecal administration of tofersen in patients with familial ALS who have *SOD1* gene mutations [78]. Fifty participants were randomized at a 3:1 ratio to receive either tofersen or placebo, with five doses administered over a 12-week period. Participants in the tofersen group received doses of 20, 40, 60, or 100 mg. The primary endpoints were safety and pharmacokinetics, while the secondary endpoint was the change from baseline in CSF SOD1 protein concentration at day 85. Forty-eight participants completed all five scheduled doses. There were three reported deaths: one in the placebo group, due to respiratory failure during the trial; one in the 20 mg group, due to pulmonary embolism during the follow-up period; and another in the 60 mg group, due to respiratory failure during the follow-up. Most patients experienced adverse events related to lumbar puncture, and increased CSF protein and white cell counts were also observed; however, tolerability was generally good and deemed safe. The difference in the change from baseline in CSF SOD1 concentration between the tofersen and the placebo groups at day 85 was −33 percentage points (95% confidence interval [CI], −47 to −16) for the 100 mg dose; a decrease in CSF SOD1 concentration was observed in the higher concentration group compared with the lower concentration group.

#### 5.3.2. Phase 3 VALOR

Following the promising results of the Phase 1/2 study, tofersen advanced to a randomized, double-blind, placebo-controlled Phase 3 study (VALOR; NCT02623699) utilizing the maximum dose from the previous trial [26]. In this Phase 3 trial, 108 patients with ALS carrying *SOD1* mutations were randomized at a 2:1 ratio to receive either 100 mg tofersen or placebo, administered intrathecally for a total of eight doses over 24 weeks. To reduce the impact of ALS heterogeneity, participants were divided into subgroups with faster and slower disease progression. The primary endpoint was the change from baseline to week 28 in the total score on the ALS Functional Rating Scale–Revised (ALSFRS-R; range, 0 to 48, with higher scores indicating better function) [79] in the fast-progressing group. Secondary endpoints included changes in total CSF SOD1 protein levels, plasma neurofilament light chain (pNFL) levels, handheld dynamometry of 16 muscles, and slow vital capacity. pNFL is a marker known to be strongly associated with death and progression of ALS [80]. Among the 60 participants in the faster-progression subgroup of the primary analysis, the change in the ALSFRS-R total score from baseline to week 28 was –6.98 points in the tofersen group and –8.14 points in the placebo group (difference, 1.2 points; 95% CI, –3.2 to 5.5; *p* = 0.97), indicating that the primary endpoint was not achieved. However, regarding some secondary endpoints, tofersen resulted in greater reductions in CSF SOD1 concentration and pNFL levels than placebo. Specifically, in the faster-progressing group, CSF SOD1 protein levels decreased by 29% in the tofersen-treated participants, while they increased by 16% in the placebo group (*p* value not reported because of the primary endpoint not being achieved). Additionally, pNFL levels decreased by 60% in the tofersen group, whereas they increased by 20% in the placebo group. Results for secondary clinical endpoints did not show significant differences between the two groups. Adverse events occurring in more than 10% of the tofersen group and more frequently than in the placebo group were pain, fatigue, arthralgia, increase in CSF white blood cells, and myalgia. Neurologic serious adverse events were reported in 7% of patients receiving tofersen, including myelitis, chemical or aseptic meningitis, lumbar radiculopathy, increased intracranial pressure, and papilledema. Thus, while the Phase III VALOR study did not achieve the primary clinical endpoint, based on biomarker responses, tofersen was approved by the FDA in April 2023 and subsequently by the Committee for Medicinal Products for Human Use (CHMP) in February 2024 [8].

#### 5.3.3. Phase 3 Extension Study

An open-label Phase 3 extension study (NCT03070119) is currently ongoing to assess the long-term safety and tolerability of tofersen. The results from this study up to 52 weeks are being integrated with the VALOR trial data, comparing participants who started tofersen at the beginning of the trial (early-start cohort) with those who switched from placebo to treatment at week 28 (delayed-start cohort) [26]. At 52 weeks, earlier initiation of tofersen was associated with less decline in functional measures, including ALSFRS-R, slow vital capacity, and handheld dynamometry, compared with the delayed start open-label extension cohort. However, the open-label extension design did not allow for direct comparison with the placebo group, limiting interpretation. The potential effects of early versus delayed initiation of tofersen will be further evaluated in the extension phase.

### 5.4. Trends after Tofersen Approval

The current protocol for tofersen includes an initial loading dose of 100 mg administered three times at 14-day intervals, followed by a maintenance dose of 100 mg once every 28 days [81]. Although the pharmacokinetics of tofersen in humans are not fully understood, autopsy studies have shown that intrathecally administered tofersen is distributed to CNS tissues. The effective half-life in the CSF is estimated to be four weeks. Tofersen is transferred from the CSF to the systemic circulation with a median time to maximum plasma concentration (Tmax) of 2 to 6 h. The elimination pathway has not been determined.

Clinical data following the approval of tofersen are beginning to be reported. In a multicenter cohort study conducted in Germany involving 24 patients who received tofersen in a real-world setting, clinical parameters, laboratory findings, and biomarkers were tracked. As in the VALOR trial and its open-label extension, a decrease in pNFL levels under treatment with tofersen was observed, along with a reduction of pNFH in CSF [82]. The therapy was deemed safe, as no persistent symptoms were noted. However, pleocytosis and immunoglobulin synthesis in CSF with clinical symptoms related to myeloradiculitis in two patients warranted careful monitoring. In a multicenter observational study involving more than 18 months of tofersen treatment, the ALS progression rate (ALS-PR), as measured by the monthly change in ALSFRS-R [83], demonstrated a mean change of 0.2 (range 0 to 1.1) and a relative reduction of 25%. Among the 16 patients, 7 showed an increase in ALSFRS-R [84]. Furthermore, patient-reported outcomes indicated a favorable perception of tofersen treatment in clinical practice.

#### Phase 3 ATLAS

Currently, a clinical trial is underway for asymptomatic *SOD1* mutation carriers (ATLAS; NCT04856982) [85]. The Phase 3 ATLAS study targets presymptomatic carriers of high- or full-penetrance *SOD1* mutations known to be associated with rapid disease progression and biomarker evidence of disease activity such as elevated pNFL levels. The trial will test the hypothesis that initiating tofersen treatment in these presymptomatic carriers will delay the onset of clinically evident ALS or reduce functional decline over time compared with initiating treatment after the onset of clinical symptoms of ALS. ATLAS is the first interventional trial in presymptomatic ALS. This randomized controlled trial plans to enroll approximately 150 presymptomatic carriers by August 2027. pNFL levels will be frequently measured, and upon detecting an increase, participants will receive either 100 mg tofersen or a placebo monthly for two years. The primary outcomes include the percentage of participants who develop clinically evident ALS within 12 and 24 months after randomization, as well as the time from randomization to the onset of clinically evident ALS. During the open-label extension phase, participants who develop clinically manifest ALS will have the opportunity to receive tofersen. The ATLAS study will deepen our understanding of ALS biomarkers and the disease’s progression, especially in the early stages, making it a valuable investigation, and its results are eagerly awaited.

Table 2 summarizes a series of clinical trials related to *SOD1* ALS.

## 6. Future Directions

Thirty years after the discovery of *SOD1* gene mutation, tofersen, an ASO for *SOD1* ALS, has finally been approved. The series of research and development efforts for treatments has not been a smooth journey, but it has significantly contributed to the understanding of the pathophysiology of ALS, providing great hope for a disease that has had almost no effective treatments to date. *SOD1* ALS, which is the target for tofersen, accounts for about 2% of all ALS cases, and it is hoped that the approval of ASOs for other mutations will be accelerated. The ASO gapmer for *FUS*-mutated ALS (jacifusen) is currently in a Phase 3 trial (FUSION; NCT04768972) and is the most realistic next ASO approved drug. ASO development for *C9orf72* mutations and several phase I/II clinical trials have been conducted, but unfortunately, no valid results have been obtained to date [71]. The establishment of the utility of intrathecal administration and the clinical trial protocol that led to FDA approval of tofersen are expected to greatly benefit future research not only in ALS but in CNS diseases and hereditary rare diseases as a whole.

The approval of tofersen is by no means the end goal; rather, it is merely a milestone. Verifying the long-term efficacy and safety of tofersen remains an important challenge ahead. The VALOR study showed improvements in biomarkers with tofersen treatment, but clinical endpoints did not demonstrate improvement, highlighting a discrepancy. Some findings suggesting clinical efficacy of tofersen are beginning to emerge in real-world settings, which is noteworthy [84,86]. pNFL levels increase 6–12 months before phenoconversion, and there may be a delay between pNFL reduction and clinical benefit [87]. This suggests that the timing of the primary endpoint evaluation in VALOR study might have been too early. In any case, it is crucial to accumulate and scrutinize data from clinical trials and real-world settings to establish evidence for the optimal use of this costly medication. The ongoing ATLAS study, which targets presymptomatic carriers, holds significant potential. If the efficacy of presymptomatic administration of tofersen is confirmed, the focus of ALS treatment and development may significantly shift towards the prevention stage.

Over 90% of rare diseases lack standard treatment options [88]. Many rare diseases are caused by pathogenic mutations in a single gene, making ASO-based therapies a promising treatment approach moving forward. Research and development of ASO therapies with greater efficacy and safety remains an important theme, especially for CNS diseases, where alternative approaches to burdensome intrathecal injections are highly desirable, and future outcomes are eagerly anticipated.

## Figures and Tables

**Figure 1 genes-15-01342-f001:**
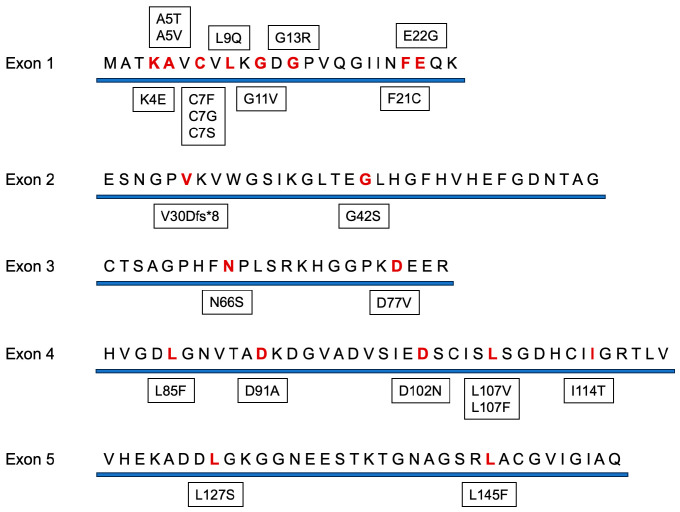
Amino acid sequence of SOD1 and representative mutations known in ALS. Mutation data were extracted from the ALS online database (ALSoD, http://alsod.iop.kcl.ac.uk, accessed on 11 October 2024). Of the more than 200 known mutations, mutations reported in at least 5 individuals are listed here. Red letters indicate amino acids with high mutation frequency.

**Figure 2 genes-15-01342-f002:**
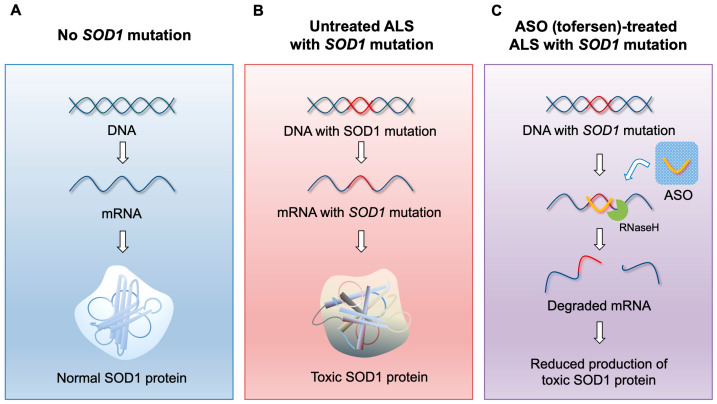
ASO (tofersen) mechanism of action. (**A**) In the absence of the *SOD1* gene mutation, functional SOD1 protein is produced by normal transcription and translation. (**B**) In the presence of *SOD1* gene mutation, toxic SOD1 protein is produced by its transcription and translation. (**C**) When tofersen, an ASO gapmer against the *SOD1* gene, specifically binds to *SOD1* mRNA, *SOD1* mRNA is degraded by the activity of RNase H, inhibiting the production of toxic SOD1 protein.

**Table 1 genes-15-01342-t001:** Summary of FDA-approved drugs for ALS.

Drug	Year	Administration	Mechanism	Note
riluzole	1995	Oral	Glutamate antagonist	Thickened riluzole was approved in 2018. Oral film was approved in 2019.
edaravone	2017	Intravenous, Oral	Free radical scavenger	Oral formation was approved in 2022.
PB/TURSO	2022	Oral	Inhibition of motor neuron apoptosis	Voluntarily removed from the market in 2024 based on the results from the Phase 3 PHOENIX trial.
tofersen	2023	Intrathecal	ASO (2′-MOE gapmer)	For ALS patients with *SOD1* mutation.

FDA, Food and Drug Administration; ALS, amyotrophic lateral sclerosis; PB/TURSO, sodium phenylbutyrate and taurursodiol; ASO, antisense oligonucleotide; 2′-MOE, 2′-O-Methoxyethyl; *SOD1*, *superoxide dismutase 1*.

**Table 2 genes-15-01342-t002:** Summary of ASO clinical trials for *SOD1* ALS.

Drug	Phase Study Type Study Objective	Number of Patients Administration, Dose	Outcome
ASO333611	Phase 1 (NCT01041222) Randomized, double-blind, placebo-controlled Safety and tolerability study of ASO333611 in *SOD1* ALS	33 patients 12 h intrathecal infusion of 0.15, 0.5, 1.5, and 3 mg	No drug-related safety issues. No reduction in SOD1 protein levels in CSF.
BIIB067 tofersen	Phase 1/2 VALOR (NCT02623699) Randomized, double-blind, placebo-controlled Safety, tolerability and pharmacokinetics study of tofersen in *SOD1* ALS	50 patients Intrathecal injection of 20, 40, 60, and 100 mg	Tofersen was generally well-tolerated and safe. The highest concentration of tofersen decreased CSF SOD1 concentrations the most.
BIIB067 tofersen	Phase 3 VALOR (NCT02623699) Randomized, double-blind, placebo-controlled Efficacy study of tofersen in *SOD1* ALS	108 patients Intrathecal injection of 100 mg	Tofersen did not improve the ALSFRS-R total score from baseline to week 28 in the faster-progression group. Tofersen resulted in a greater reduction in CSF SOD1 and pNFL concentrations.
BIIB067 tofersen	Phase 3 (NCT03070119) Open-label extension Long-term evaluation of tofersen in *SOD1* ALS	138 patients Intrathecal injection of 100 mg	Ongoing
BIIB067 tofersen	Phase 3 ATLAS (NCT04856982) Randomized, double-blind, placebo-controlled and subsequent open-label extension Long-term efficacy study of tofersen in presymptomatic carriers of *SOD1*	150 patients (estimated) Intrathecal injection of 100 mg	Ongoing

ASO, antisense oligonucleotide; *SOD1*, *superoxide dismutase 1*; ALS, amyotrophic lateral sclerosis; CSF, cerebrospinal fluid; ALSFRS-R, ALS Functional Rating Scale–Revised.

## Data Availability

No new data were created or analyzed in this study. Data sharing is not applicable to this article.

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
