# Peer review of "Recent Progress of Antisense Oligonucleotide Therapy for Superoxide-Dismutase-1-Mutated Amyotrophic Lateral Sclerosis: Focus on Tofersen"

_genes, 2024, doi:10.3390/genes15101342_

Round 1
Reviewer 1 Report
Comments and Suggestions for Authors
My suggestions:
1. In the ALS overview authors may mention the genetic overlap between ALS and FTD.
2. The authors may add a figure for chapter 2.1 on the structure of SOD1 and the mutations described in ALS cases. Furthermore, the authors may highlight the mutations that were targeted by ASO.
3. Were any side effects of drugs, used for ALS?
4. Are there ongoing animal experiments for ASO therapies for SOD1 mutations? The authors may discuss it a little more in detail.
5. Were there any side effects reported in the patients, who were treated by ASO?
Reviewer 2 Report
Comments and Suggestions for Authors
This paper represents an excellent review of the initial controlled clinical trials of the first successful anti-sense oligonucleotide (ASO), Tofersen, given to patients who had a mutation in the (cytoplasmic) superoxide dismutase protein SOD1. As the authors point out, the clinical trial path of Tofersen has been "bumpy", including obtaining FDA approval in the US based solely on biomarker proteins' improvements. (the ALSFRS-R scores did not show significant improvement in the treated cohort; thus primary endpoint was not reached).
The authors carefully and comprehensively review the projected mechanism of action of Tofersen (Figure 1 is excellent!), its limitations (mainly poor blood-brain barrier (BBB) penetration), reasonable side effect/safety profile and effects on ALS clinically (ALSFRS-R) and biomarker proteins. As suggested in this review, Tofersen represents a modest success story in monogenic ALS, and this approach may ultimately be tried in selected sporadic ALS patients. ASO's against other mutated genes in ALS are being developed, and carrier molecules to improve BBB penetration are also being developed. After decades of therapeutic "desert", the treatment of ALS is finally yielding to scientific improvements brought by molecular genetics.
Tables 1 and 2 are contributory and helpful.
I enjoyed reading this review and hope the authors consider comparable reviews of other ASO treatments for genetic and sporadic ALS. I feel this review has been carefully written and represents a valuable addition to the medical-scientific literature.
Reviewer 3 Report
Comments and Suggestions for Authors
The authors summarize recent progress on the ASO therapy against ALS with tofersen. This is an interesting topic to work on. The review is well written and organized.
some minor defects:
1, the authors should also make some discussion on progression of other ASOs used for ALS therapy, such as targeting FUS protein
2, the authors should discuss briefly on how tofersen is metabolized in human body, distributed after therapy in human body and in neurons.
Round 2
Reviewer 1 Report
Comments and Suggestions for Authors
The authors fulfilled my suggestions. Thank you.